# Poor Health Behaviour in Medical Students at a South African University: A Cross-Sectional Survey Study

**DOI:** 10.3390/ijerph21070824

**Published:** 2024-06-24

**Authors:** Bert Celie, Ria Laubscher, Martin Bac, Marianne Schwellnus, Kim Nolte, Paola Wood, Tanya Camacho, Debashis Basu, Jill Borresen

**Affiliations:** 1Unité de Recherche en Physiologie Cardio-Respiratoire, Faculté des Sciences de la Motricité, Université Libre de Bruxelles, 1070 Bruxelles, Belgium; 2Sport, Exercise Medicine and Lifestyle Institute (SEMLI), Faculty of Health Sciences, University of Pretoria, Pretoria 0002, South Africa; kimnolte1@gmail.com (K.N.); paola.wood@up.ac.za (P.W.); tccamacho4@gmail.com (T.C.); jill.borresen@up.ac.za (J.B.); 3IOC Research Centre of South Africa, Pretoria 0002, South Africa; 4Biostatistics Unit, South African Medical Research Council, Cape Town 7505, South Africa; ria.laubscher@mrc.ac.za; 5Department of Family Medicine, Faculty of Health Sciences, University of Pretoria, Pretoria 0002, South Africamarianneschwellnus@gmail.com (M.S.); 6Department of Physiology, Division of Biokinetics and Sports Science, Faculty of Health Sciences, University of Pretoria, Pretoria 0002, South Africa; 7Department of Public Health Medicine, Faculty of Health Sciences, University of Pretoria, Pretoria 0002, South Africa; debashis.basu@up.ac.za

**Keywords:** non-communicable diseases, behavioural risk factors, medical students

## Abstract

Background: Personal health behaviours and lifestyle habits of health professionals influence their counselling practices related to non-communicable diseases (NCDs). There are limited data on the prevalence of unhealthy lifestyle habits among medical students and the impact of acquired health knowledge throughout the curriculum. The aim of this study was to determine and compare the prevalence of modifiable behavioural NCD risk factors of medical students in different academic years at a South African tertiary institution. Methods: A cross-sectional observational study of 532 consenting medical students was conducted. Participants completed five online questionnaires regarding lifestyle behaviours (physical activity, dietary habits, smoking, alcohol consumption and sleep). Results: Lifestyle-related risk factors with the highest prevalence were poor sleep quality (66.0%), low levels of habitual physical activity (55.8%) and low-to-moderate diet quality (54.5%). There were no differences between academic years for all risk factors measured. Over 60% of the cohort had two or more NCD risk factors and this prevalence did not differ across the degree program with the acquisition of more health knowledge. Conclusion: Medical students have a high prevalence of poor sleep quality, low levels of physical activity and low-to-moderate diet quality, which does not appear to change over the course of their academic career. Sleep hygiene, regular physical activity and healthy nutrition should be targeted in intervention programmes and be more prevalent in the medical curriculum.

## 1. Introduction

Personal health behaviours of health professionals have been presented as important determinants of both their counselling practices and community health behaviour and health in general [1,2,3,4,5,6]. For example, non-smoking medical doctors who engage in regular physical activity and have good nutritional habits and a healthy body composition are more likely to prescribe lifestyle behavioural changes and are perceived as more credible by their patients in general [1,2,3,4,5,6]. A healthy lifestyle behaviour has been repeatedly mentioned as the most important countermeasure in order to prevent non-communicable diseases (NCDs), and for that reason, it is crucial to have medical doctors on board, as they are considered as the representatives of ’health’ by their direct environment [7,8,9]. NCDs are the leading cause of death and morbidity globally, being responsible for approximately 70% of all deaths [7,8,9]. While the World Health Organisation (WHO) has attributed 51% of South African deaths to NCDs, it is clear that people of all age groups, regions and countries are vulnerable to the risk factors contributing to NCDs. Physical inactivity, unhealthy diet, tobacco use and the harmful use of alcohol are regarded as the most important modifiable behavioural contributors to the development of NCDs [7,8,9]. Additionally, a considerable body of epidemiological evidence adds poor sleep to that list of crucial behavioural NCD risk factors [10,11]. While there is an abundance of literature focusing on the prevalence of a single behavioural non-communicable disease (NCD) risk factor in (medical) students, data about combined NCD risk factors is surprisingly scarce, especially from the African continent [12,13,14,15,16,17]. Taking into account the important impact of the health professionals’ (health) behaviour for NCD treatment and prevention in the broader community and the scarcity of data on this issue in the South African context, it is crucial to investigate lifestyle habits during the student period as it seems that these habits, established in the university student period, will determine health behaviour later in life and significantly improve academic performance as well as NCD prevention and treatment in the broader community [12,13,14,15,16]. Considering the importance of the latter, there is an urgent need to gain more knowledge about the prevalence of NCD risk factors among future health professionals. In a preliminary study by this research group, we found that a concerningly large proportion of first-year medical students self-reported an unhealthy lifestyle with poor sleep quality, physical inactivity and unhealthy diet as the most prominent NCD risk factors [18]. The question that remained unanswered, however, was whether the attainment of health knowledge through their academic career would improve health behaviour.

In March 2020, the South African government proclaimed one of the hardest COVID-19 related lockdowns worldwide, with a strict confinement of the population to their residences. With an incremental attenuation of this lockdown from level 5 to level 1, allowing more freedom of movement outside the home, home confinement regulations were finally lifted entirely in August 2020. An Australian study has already shown that physical activity, sleep, alcohol consumption and smoking were negatively impacted by these regulations in a similar lockdown context [19]. However, inconclusive or no results in (medical) student cohorts have been reported in studies showing both better and worse sleep quality, physical (in)activity, nutritional quality and poorer quality of life [20,21,22].

The primary aim of this study was to determine the prevalence of modifiable behavioural NCD risk factors in medical students at a South African university and whether there were cross-sectional differences between different years throughout the academic program. The secondary aim was to determine the number of medical students that possessed one or multiple modifiable behavioural risk factors. The possible impact of COVID-19 lockdown regulations on health behaviours is taken into consideration with the interpretation of the data.

## 2. Materials and Methods

### 2.1. Participants and Procedures

In a cross-sectional observational study conducted in 2020, medical students of 18 years or older and currently registered for a Bachelor of Medicine and Surgery (MBChB, a 6-year medical degree) at the Faculty of Health Sciences, University of Pretoria, were invited to complete five online questionnaires. After an information session during which the research project and procedures were explained to all potential participants, 532 medical students from the 1st-, 2nd-, 3rd- and 5th-year cohorts (44.6% of the invited sample) gave their consent and completed five online questionnaires via the online Smartabase platform/application (Fusion Sports Pty (Ltd.), Teamsports, Durham, North Carolina, USA). Also, it was communicated to all students that no consequences on their examinations and grading would follow in case students did not reply to the surveys. The cohort that completed all questionnaires consisted out of 356 females (66.9%) and 176 males (33.1%), and the age range of our sample is between 20 and 24 years old. Questionnaires focused on lifestyle behaviours including smoking, alcohol consumption, habitual physical activity, nutritional habits and sleep quality. The 4th-year students did not consent to participate in this study. Ethics approval was obtained from the Health Sciences Research Ethics Committee at the University of Pretoria (REC no: 104/2019).

### 2.2. Measurement of Modifiable Behavioural Risk Factors

Tobacco use. Tobacco use was assessed using relevant sub-sections of the WHO Global Adult Tobacco Survey (GATS), which monitors tobacco use among adults of 15 years and older. Outcome measures included daily smoking and past smoking [23].

Alcohol consumption. The WHO Alcohol Use Disorder Identification Test (AUDIT) is a ten-item questionnaire used to screen adults for harmful alcohol consumption [24]. Total AUDIT scores range from 0 to 40, and scores higher than eight are considered indicative of the harmful use of alcohol.

Physical activity. The Kasari FIT index was used to assess levels of habitual physical activity and was calculated from responses on the activity questionnaire scale (FIT = Frequency (F) × Intensity (I) × Time (T)). Total Kasari FIT Index scores range between 1 (physically inactive) and 100 (very physically active). A total FIT Index score lower than 36 is indicative of low habitual physical activity [25,26].

Nutrition. The short version Rapid Eating Assessment for Participants (REAP-S) is a questionnaire that has been widely used to assess diet quality in daily life [27]. Possible total REAP-S scores range from 13 to 39 with a higher score indicating a higher diet quality. Low, moderate and good diet quality scores were defined as 15–19, 20–29 and 30–39, respectively [28].

Sleep quality. The Pittsburgh Sleep Quality Index (PSQI) is a 19-item survey which assesses seven subscales of sleep quality, such as subjective sleep quality, sleep latency, sleep duration, habitual sleep efficiency, sleep disturbances, use of sleeping medication and daytime dysfunction [29,30]. The combined scores of the subscales determine the global sleep quality score with higher scores indicative of poorer sleep quality and lower scores indicating better or good sleep quality. A global score of 5 has been established as a clinical cut-off for the determination of poor sleep [29,30].

### 2.3. Statistical Analysis

Statistical analyses were performed using the STATA V16 statistical package. Data was collected via validated and reliable online questionnaires from consenting students and extracted from Smartabase to Microsoft Excel for further analysis. The normality of the continuous variables was checked using the Tukey ladder of powers. This method allows one to change the shape of a skewed distribution so that it becomes normal or nearly normal. After a rigorous check, all continuous variables fit normal distributions. A one-way ANOVA was used to detect cross-sectional differences between students of different academic years, and the Chi square test was used to assess differences between years for categorical modifiable behavioural risk factors. The significance level was set at *p*-values less than 0.05. The Levene’s test for homogeneity of variance was not significant for any modifiable risk factor except sleep quality, indicating that there was no significant difference in variances between the different years except for sleep quality.

## 3. Results

In Table 1, the average total scores for questionnaires (where a score is calculated), are depicted for all the students and for the different academic years. Although no significant differences were observed between the academic years for these health behavioural risk factors, it seems that physical activity increased slightly over the degree program (*p* = 0.219). On average, the Total Kasari FIT Index score for all students and for each year was slightly above the cut-off criterion of <36 (indicating low habitual physical activity); thus, on average, the student base reported moderate levels of physical activity, although the variance around these averages was large. The total PSQI score for all students and for each academic year was consistently above the cut-off criterion of 5 or more, indicating overall poor sleep quality. Also, the average diet quality was in the moderate range of 20–29 for all students and for each academic year, except the 5th year. In the last year of their academic program, the nutritional quality was slightly higher than 29. The average AUDIT score was considerably lower (<8), which is indicative of a good average attitude towards alcohol in the current cohort.

In Table 2, the prevalence of NCD behavioural risk factors is presented for all students and for the different years. No significant differences were detected in these health behaviours between academic years, and high prevalence ratios were found for poor sleep quality, low levels of habitual physical activity and low-to-moderate diet quality across the different stages of the degree program. Indicators of smoking status and alcohol consumption were excellent, with prevalences ranging between 2.1% and 6.9% for all students and each academic year. The low to moderate diet quality seemed to decrease slightly during the medical education program; however, no significant difference between academic years was observed.

In Table 3, the prevalence of students with no, one and multiple risk factors is presented. Of the medical students that participated in this study, a staggering proportion of 62.0% had two or more behavioural risk factors. Surprisingly, only 11.7% of the student sample did not possess any behavioural risk factor, while 26.3% of this cohort possessed only one risk factor. There were no significant differences in the clustering of behavioural risk factors between students in different academic years of the degree program (*p* = 0.460).

## 4. Discussion

The first finding was a surprisingly high prevalence of behavioural NCD-related risk factors in medical students through all years of the medical degree program, with poor sleep quality (66.0%), physical inactivity (55.8%) and low to moderate diet quality (54.5%) being the most predominant risk factors. The prevalence of these risk factors did not differ significantly between the different academic years. Second, over 60% of the medical students were found to have two or more of these modifiable risk factors, and the prevalence of these clustered risk factors (two or more) was not significantly different between students in different years of their academic career.

Despite augmented health literacy, our cohort of medical students was found to possess an unhealthy lifestyle across all years in a South African medical degree program. Moreover, the acquired health literacy did not proportionally change the students’ health behaviour throughout the medical program, as no significant differences were found between students in different years. This concerning fact could be due to a lack of preventive or lifestyle healthcare courses or to a lack of time to prepare healthy food, be physically active and sleep well throughout the medical curriculum. These results are in agreement with a recent Italian study that presented either analogous or worse health behaviours through a comparable academic program [31]. While a higher proportion of students with multiple NCD risk factors was observed in this current investigation compared to Faggiano et al. [31], it seems that substance use (alcohol and tobacco) was better compared to the Italian medical student sample. Additionally, a similar proportion of students was found to have a poor level of daily physical activity in the South African and Italian medical cohorts. The latter study, however, found their cohort to have significantly better nutritional behaviours in the senior years, which was not found in our study. It is difficult to compare our findings with other scientific data because other studies either focused on one behavioural risk factor or a certain academic year, according to a recently published meta-analysis [17]. The results from our cohort of medical students are unexpected and concerning given their attainment of health knowledge through thorough medical training and given that the lifestyles of health professionals have been shown to be crucial for the broader community’s health [6,7,8,9,10,11].

Two-thirds of medical students attained PSQI scores that indicated poor sleep quality, which is higher than the pooled prevalence of 52.7% reported in a meta-analytical dataset including 25,735 medical students worldwide [32]. The meta-analytical data showed that a higher prevalence of poor sleep quality was associated with or biased by smaller sample sizes [32]. However, if the sample size of the current study is compared to the manuscripts included in the meta-analysis, only 5 of the included 57 studies had more participants, placing this study in the top 10 percent with respect to sample size. Hence, it is unlikely that low sample sizes introduced a bias into the results of the current study. An alternative explanation might be related to the African region being under-represented in the above-mentioned meta-analysis (2 Nigerian, 1 Egyptian and 1 Sudanese study) [32]. Multiple studies have shown that poor sleep quality is significantly associated with raised stress levels and anxiety in medical students [33,34]. High academic expectations are mainly responsible for these elevated stress levels, but students could be impacted by a vicious cycle of stressors that negatively affect sleep quality, inducing even more stress [33].

We can only speculate about the specific impact of the COVID-19 pandemic and its related lockdown regulations on the general sleep quality of medical students—a population reportedly already at risk for elevated emotional distress and suicidal ideation [33,34]. In the general population, it has been shown that poor sleep quality significantly increased and was associated with higher depression, anxiety and stress symptoms during COVID-19 lockdowns [19]. A possible explanation for these findings could be the disruption of the circadian sleep rhythm during home confinement [35]. For example, a recent Canadian study found that half of a student cohort felt more depressed and lonelier during COVID-19 lockdowns, with detrimental effects on their stress levels [20]. While the latter investigation did not study sleep quality, it could be hypothesized that augmented stress and depression levels as well as an adjusted circadian rhythm would have a serious impact on students’ sleep quality. Although mixed results were reported with either better or worse sleep quality among Chinese and Turkish medical students during the COVID-19 lockdown, the severity of lockdown regulations could have major implications on health behaviour in general [21,36]. As South Africa implemented one of the strictest home confinement regulations globally in March 2020, it can be speculated that this may have contributed to the surprisingly high prevalence of poor sleep quality among other poor health behavioural parameters in our study.

The second most prevalent risk factor found in our medical student cohort was insufficient habitual physical activity, with 55.8% reporting low levels. This is slightly lower than a previous study from this research group that reported a prevalence of 64.1% in 1st-year medical students before the COVID-19 pandemic [18], and it is lower than other studies investigating this issue in similar cohorts [31,37]. The third most prevalent risk factor was low-to-moderate diet quality (54.5%) which was also lower in this (2020) cohort compared to the 2019 cohort [18]. The poor diet quality and physical activity habits among medical students could be due to the challenge of living independently for the first time without parental constraints, higher screen time, difficult access to healthy food, social drinking and less time to be physically active [38,39]. It is, however, very concerning to observe these results in future health professionals, because medical doctors with proper nutritional habits who engage regularly in physical activity are more likely to prescribe lifestyle behavioural changes and are perceived as more credible by their patients in general [6,9,10,11]. Inadequate exercise prescription has been observed in USA-based research despite evidence showing that physically active doctors are more likely to prescribe physical activity in their practice [40,41,42]. Moreover, regular physical activity has been recommended by several international organizations as a crucial cornerstone in NCD prevention and treatment, with a wide range of health and economic benefits [1,3,43]. Both nutrition and physical activity could have been severely impacted by the South African COVID-19 lockdown regulations, where physical activity outside the house was completely prohibited in the first phase and only allowed during certain time slots from 1 May 2020. Recent studies from other countries have found that physical activity was negatively impacted by the lockdown regulations in general; however, opposing results have been reported on nutritional quality in different studies investigating the matter [20,21]. Longitudinal research about changes in physical activity during different stages of lockdown life has shown that young adults are more likely to be in a ‘dynamic group’ with either a decreasing or increasing amount of physical activity during different stages of lockdown [44]. It has also been suggested that in the case where younger people are better educated, the chances are higher that physical activity during lockdown will increase [44]. Hence, it can be hypothesized that physical activity in medical students improved slightly during the different stages of home confinement compared to our previous study conducted in 2019, before COVID-19 [18].

A first limitation of this study is that response ratios differed significantly between the different years. Although normality was checked and preserved throughout all analyses, this might have had an impact on the analyses of variance between the different years. Also, it was unexpected that COVID-19 would occur and have an impact on the lifestyles and behaviours of medical students. Hence, it was important to take this into account when analysing the results of this study, which could act as both a strength and a limitation of this study. A second limitation was the absence of data on 4th-year students. A third limitation was the self-reported nature of the retrieved data in the current study, which has the potential disadvantage of over- or under-representing certain lifestyle behaviours. An important strength of this study is the large total sample size of medical students that completed all questionnaires, which provides reliable data with regard to the health behaviour of the total sample.

## 5. Conclusions

In conclusion, we found a high prevalence of unhealthy lifestyle behaviours in a medical student cohort with regard to physical activity, nutrition and sleep quality that does not seem to improve through the course of an academic degree program with increasing knowledge about general health. Moreover, a considerable number of the students reported having two or more behavioural risk factors, putting them at increased risk for CVD. Taking into consideration the societal importance of future medical doctors for health in general, these findings are concerning and should be targeted in educational programs in order to strengthen preventive approaches in public health.

Based on these results, we recommend that future interventions implemented at tertiary institutions could include the following: 1. Strengthening the medical and allied health curricula to include more education about NCDs and their risk factors and prevention, including areas such as behaviour change theory to provide a deeper understanding of the topic. 2. Reimagining the built environment to facilitate changes in behaviour, e.g., healthy food options at various canteens/cafeteria; indoor/outdoor exercise facilities available to students; safe outdoor spaces/green corridors to encourage walking and meeting on campus; and building design that encourages physical activity. 3. Campus-wide health promotion initiatives including communications, posters and activations to create awareness. Taking into consideration the widespread effects of NCDs on health, future research should focus on evaluating these modifiable risk factors longitudinally before and after certain implemented initiatives during the academic program, such as an augmented amount of specific and NCD-prevention-oriented courses, some movement classes or active breaks and the availability of affordable, healthy food for students.

## Figures and Tables

**Table 1 ijerph-21-00824-t001:** Average health behaviour profiles (means ± SD) of first-, second-, third- and fifth-year medical students.

	All Participants	1st Year (n = 187)	2nd Year (n = 232)	3rd Year (n = 43)	5th Year (n = 70)	*p*-Values
Mean ± SD						
Sleep quality (PSQI score)	5.92 ± 2.93	5.74 ± 2.73	6.09 ± 2.86	6.05 ± 2.76	5.74 ± 3.69	0.624
Kasari FIT Index score	38.73 ± 27.37	37.33 ± 27.45	37.58 ± 27.73	41.88 ± 26.02	44.36 ± 26.47	0.219
REAP-S score	28.68 ± 4.46	28.73 ± 4.19	28.43 ± 4.61	28.98 ± 5.09	29.23 ± 4.31	0.576
AUDIT score	1.87 ± 2.40	1.63 ± 2.07	2.05 ± 2.76	1.81 ± 2.10	1.93 ± 2.11	0.343

* significant difference (*p* < 0.05).

**Table 2 ijerph-21-00824-t002:** Prevalence (%) of modifiable NCD-related risk factors in 1st-year, 2nd-year, 3rd-year and 5th-year medical students.

	All		1st Years		2nd Years		3rd Years		5th Years		*p*-Values
	N	%	N	%	N	%	N	%	N	%	
Past or current smoking	29	5.5	7	3.7	16	6.9	2	4.7	4	5.7	0.560
Harmful alcohol use (AUDIT score ≥ 8)	22	4.1	4	2.1	13	5.6	1	2.3	4	5.7	0.260
Low physical activity (FIT index ≤ 36)	297	55.8	110	58.8	131	56.5	21	48.8	35	50.0	0.469
Moderate physical activity (FIT index >36–<64)	103	19.4	36	19.3	42	18.1	11	25.6	14	20.0	0.724
Low frequency (<3 times/week)	128	24.1	43	23.0	53	22.8	14	32.6	18	25.7	0.550
Low to moderate intensity	290	54.5	107	57.2	128	55.2	20	46.5	35	50.0	0.517
Time (<30 min/session)	267	50.2	99	52.9	120	51.7	17	39.5	31	44.3	0.294
Low-to-moderate diet quality (REAP-S score ≤ 29)	290	54.5	103	55.1	134	57.8	22	51.2	31	44.3	0.245
Low diet quality (13–19)	17	3.2	5	2.7	9	3.9	2	4.7	1	1.4	0.679
Moderate diet quality (20–29)	273	51.3	98	52.4	125	53.9	20	46.5	30	42.9	0.376
Poor sleep quality (PSQI score > 5)	351	66.0	125	66.8	157	67.8	30	69.8	39	55.8	0.270

* significant difference (*p* < 0.05).

**Table 3 ijerph-21-00824-t003:** Prevalence (%) of multiple modifiable NCD-related risk factors in 1st-year, 2nd-year, 3rd-year and 5th-year medical students. (*p* = 0.460).

	1st Years		2nd Years		3rd Years		5th Years		All	
	N	%	N	%	N	%	N	%	N	%
No risk factors	24	12.8	21	9.1	5	11.6	12	17.1	62	11.7
One risk factor	43	23.0	64	27.6	13	30.2	20	28.6	140	26.3
Two or more risk factors	120	64.2	147	63.4	25	58.1	38	54.3	330	62.0

## Data Availability

Data are available from the corresponding author (Bert Celie, bert.celie@ulb.be) upon reasonable request and signed access agreement.

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
