# Peer review of "Poor Health Behaviour in Medical Students at a South African University: A Cross-Sectional Survey Study"

_ijerph, 2024, doi:10.3390/ijerph21070824_

Round 1

Reviewer 1 Report

Comments and Suggestions for Authors
  • Contextualization: Clearly state why understanding the lifestyle habits of medical students specifically is important, especially in the context of South Africa. How might these habits affect their future roles as healthcare providers? Expand on the link between personal health behaviors of health professionals and their effectiveness in counseling on NCDs.

Methods

  • Study Design Clarification: You mention the use of five online questionnaires. Please specify the origin or validation status of these questionnaires to establish their reliability and appropriateness for this study.
  • Sample Description: Provide details about the demographics of the study participants (e.g., age range, gender distribution, year distribution). This information is crucial for understanding the representativeness of the sample.
  • Statistical Analysis: The methods mention using a Chi-square test and one-way ANOVA. Clarify whether the correct assumptions for these tests have been met (e.g., independence, normality, homogeneity of variance). Also, consider mentioning how you handled potential confounders or covariates.

Results

  • Detailed Findings: When presenting your results, ensure that you clearly display the prevalence of each risk factor by academic year. Tables or graphs could be more effective here.
  • Comparative Analysis: Highlight any significant differences found between academic years. Discuss any trends that emerge, such as increases or decreases in risk factors as students progress through their education.

Discussion

  • Interpretation of Results: Discuss what the observed prevalence rates and differences might imply about the impact of medical education on lifestyle habits.
  • Contextual Relevance: Compare these findings with existing literature on medical students in other regions or countries. This can help to contextualize your results and might suggest areas where the curriculum could be improved to address these risk factors.
  • Limitations: Address potential biases or limitations in your study, such as the self-reported nature of the data, which may lead to underreporting or overreporting of certain behaviors.

Conclusion

  • Implications for Medical Education: Based on your findings, recommend specific changes or interventions in the medical curriculum that could help in cultivating healthier lifestyle habits among students.
  • Future Research: Suggest areas for further research, such as longitudinal studies to track changes in student behaviors over time or intervention studies to test specific changes in curriculum or support services.

Comments on the Quality of English Language

moderate revision on language.

Author Response

Response to the reviewer (Reviewer 1)

  • Overall response to the reviewer: The author team wants to thank the reviewer sincerely for taking the time to read through the manuscript thoroughly and giving interesting, specific advices or detect some mistakes that were still present in the text. We are aware that this takes a lot of time and appreciate the effort. We hope the level of the manuscript increased and is now acceptable for the reviewer.

Specific comment responses

  1. Contextualization: Clearly state why understanding the lifestyle habits of medical students specifically is important, especially in the context of South Africa. How might these habits affect their future roles as healthcare providers? Expand on the link between personal health behaviors of health professionals and their effectiveness in counseling on NCDs.

  • Response to the reviewers’ question/remark: We agree with the reviewers’ opinion that this context was not yet clearly provided. Hence, we have tried to sharper define the rationale behind this study and the importance of investigating this issue. Also, we have now changed the introduction part starting with a state of the art about the importance of personal health behavior in health professionals which is then followed with some specifics about NCDs in general. The updated introduction section could be found below.

  • Adjusted text/table/figure: Personal health behaviours of health professionals have been presented as important determinants on both their counselling practices and the community health behaviour and health in general [1-6]. For example, non-smoking medical doctors who engage in regular physical activity, have good nutritional habits and a healthy body composition are more likely to prescribe lifestyle behavioural changes and are perceived as more credible by their patients in general [1-6]. A healthy lifestyle behaviour has been repeatedly mentioned as the most important countermeasure in order to prevent Non-Communicable Diseases (NCDs) and for that reason, it is crucial to have medical doctors on board as they are considered as the representatives of ’health’ by their direct environment [7-9]. NCDs are the leading cause of death and morbidity globally, being responsible for approximately 70% of all deaths[7-9]. While the World Health Organisation (WHO) has attributed 51% of South African deaths to NCDs, it is clear that people of all age groups, regions and countries are vulnerable to the risk factors contributing to NCDs. Physical inactivity, unhealthy diet, tobacco use and the harmful use of alcohol, are regarded as the most important modifiable behavioural contributors to the development of NCDs[7-9]. Additionally, a considerable body of epidemiological evidence adds poor sleep to that list of crucial behavioural NCD risk factors[10,11].While there is an abundance of literature focusing on the prevalence of a single behavioural Non-Communicable Disease (NCD) risk factor in (medical) students, data about combined NCD risk factors is surprisingly scarce, especially from the African continent [12-17]. Taking into account the important impact of the health professionals’ (health) behaviour for NCD treatment and prevention of the broader community and the scarcity of data on this issue in the South African context, it is crucial to investigate lifestyle habits during the student period as it seems that these habits, established in the university student period, will determine health behaviour later in life and significantly improve academic performance as well as NCD prevention and treatment of the broader community [12-16].

  1. Study Design Clarification: You mention the use of five online questionnaires. Please specify the origin or validation status of these questionnaires to establish their reliability and appropriateness for this study.

  • Response to the reviewers’ question/remark: Thank you for this remark. Considering the fact that only validated (and reliable) questionnaires were used in the current investigation and the relevant references showing their respective validity were already included, we adjusted the paper text slightly to meet the reviewer’s question and to convince eventual readers that validated tools were used to obtain these results. By doing so, we hope more specificity of the validation status is provided in the paper.

  • Adjusted text/table/figure: Tobacco use. Tobacco use was assessed using relevant and sub-sections of the WHO Global Adult Tobacco Survey (GATS) that is a valid and reliable tool to monitor tobacco use among adults of 15 years and older. Outcome measures included daily smoking and past smoking[23].

Alcohol consumption. The valid WHO Alcohol Use Disorder Identification Test (AUDIT) is a ten-item questionnaire, used to screen adults for harmful alcohol consumption[24]. Total AUDIT scores range from 0 to 40 and scores higher than eight are considered indicative of the harmful use of alcohol.

Physical activity. The validated Kasari FIT index was used to assess levels of habitual physical activity and was calculated from responses on the activity questionnaire scale (FIT = Frequency (F) x Intensity (I) x Time (T)). Total Kasari FIT Index scores range between 1 (physically inactive) to 100 (very physically active). A total FIT Index score lower than 36 is indicative of low habitual physical activity[25,26].

Nutrition. The short version Rapid Eating Assessment for Participants (REAP-S) is a valid and reliable questionnaire that has been widely used to assess diet quality in daily life[27]. Possible total REAP-S scores range from 13 to 39 with a higher score indicating a higher diet quality. Low, moderate and good diet quality scores were defined as 15–19, 20-29 and 30–39 respectively[28].

Sleep quality. The Pittsburgh Sleep Quality Index (PSQI) is a valid and reliable 19-item survey which assesses seven subscales of sleep quality, such as subjective sleep quality, sleep latency, sleep duration, habitual sleep efficiency, sleep disturbances, use of sleeping medication, and daytime dysfunction[29,30]. The combined scores of the subscales determine the global sleep quality score with higher scores indicative of poorer sleep quality and lower scores indicating better or good sleep quality. A global score of 5 has been established as a clinical cut-off for the determination of poor sleep [29,30].

  1. Sample Description: Provide details about the demographics of the study participants (e.g., age range, gender distribution, year distribution). This information is crucial for understanding the representativeness of the sample.

  • Response to the reviewer: We agree with the reviewers’ opinion and for that reason, we included more demographic details of the participants into the METHODS section. For example, our sample consisted out of 356 females (66.9%) and 176 males (33.1%), The age range of our sample is between 20-24 years old and we had 187, 232, 43 and 70 students that completed the questionnaires from the 1st, 2nd, 3rd, and 5th year respectively. We did not include all these details in the manuscript, however, we hope it will be sufficient to meet the reviewers’ requirements.

  • Adjusted text/table/figure: 1. Participants and Procedures

In a cross-sectional observational study conducted in 2020, medical students of  18 years or older and currently registered for a Bachelor of Medicine and Surgery (MBChB, a 6-year medical degree) at the Faculty of Health Sciences, University of Pretoria, were invited to complete five online questionnaires. 532 medical students from the 1st,2nd, 3rd and 5th year (44.6% of the invited sample) gave their consent and completed five online questionnaires via the online Smartabase platform/application (Fusion Sports Pty(Ltd)). The cohort that completed all questionnaires consisted out of 356 females (66.9%) and 176 males (33.1%) and the age range of our sample is between 20-24 years old.

  1. Statistical Analysis: The methods mention using a Chi-square test and one-way ANOVA. Clarify whether the correct assumptions for these tests have been met (e.g., independence, normality, homogeneity of variance). Also, consider mentioning how you handled potential confounders or covariates.

  • Response to the reviewer: We agree with the reviewers’ opinion that these specific details are preferably included into the manuscript. For that reason, we expanded this specific part in the manuscript with more explanation about the statistical assumptions being carried out. With regard to normality, however, we believe we included sufficient explanation already in the methods section (V130: Normality of the continuous variables were checked using the Tukey ladder of powers. This method allows one to change the shape of a skewed distribution so that it becomes normal or nearly-normal. After a rigorous check, all continuous variables fitted normality). We hope the adjustments made are in line with the reviewers’ expectations.

  • Adjusted text/table/figure: The Levene’s test for homogeneity of variance was not significant for all modifiable risk factors except sleep quality, indicating that there was no significant difference in variances between the different years except for sleep quality.

  1. Detailed Findings: When presenting your results, ensure that you clearly display the prevalence of each risk factor by academic year. Tables or graphs could be more effective here.

  • Response to the reviewer: We thank the reviewer for his effort to improve this manuscript, however, in this case we believe we have a table which is sufficient to present the requested findings. In this table the prevalences (In percent as well as the amount of students) of all risk factors and significance level are presented per academic year as well as the total prevalence of all risk factors. If the reviewer wants to have a specific change in the presentation of these data, we are, of course, willing to adhere to the requested advice of the reviewer. We have attached the table as it appears in the manuscript below.

  • Not-adjusted text/table/figure:

 Table 2. Prevalence (%) of modifiable NCD-related risk factors in 1st year, 2nd year, 3rd year and 5th year medical students.

All

1st years

2nd years

3rd years

5th years

p-values

N

%

N

%

N

%

N

%

N

%

Past or current smoking

29

5.5

7

3.7

16

6.9

2

4.7

4

5.7

0.560

Harmful alcohol use (AUDIT score >=8)

22

4.1

4

2.1

13

5.6

1

2.3

4

5.7

0.260

Low physical activity (FIT index <=36)

297

55.8

110

58.8

131

56.5

21

48.8

35

50.0

0.469

Moderate physical activity (FIT index >36 - <64)

103

19.4

36

19.3

42

18.1

11

25.6

14

20.0

0.724

Low Frequency (<3 times/week)

128

24.1

43

23.0

53

22.8

14

32.6

18

25.7

0.550

Low to moderate intensity

290

54.5

107

57.2

128

55.2

20

46.5

35

50.0

0.517

Time (<30 minutes/ session)

267

50.2

99

52.9

120

51.7

17

39.5

31

44.3

0.294

Low to moderate diet quality (REAP-S score <=29)

290

54.5

103

55.1

134

57.8

22

51.2

31

44.3

0.245

Low diet quality (13-19)

17

3.2

5

2.7

9

3.9

2

4.7

1

1.4

0.679

Moderate diet quality (20-29)

273

51.3

98

52.4

125

53.9

20

46.5

30

42.9

0.376

Poor sleep quality (PSQI score >5)

351

66.0

125

66.8

157

67.8

30

69.8

39

55.8

0.270

* significant difference (p<0.05).

  1. Comparative Analysis: Highlight any significant differences found between academic years. Discuss any trends that emerge, such as increases or decreases in risk factors as students progress through their education.

  • Response to the reviewer: Despite the fact that we already included a small paragraph discussing the eventual risk factor differences and trends throughout the academic curriculum, we have enlarged the paragraph and elaborated a bit more on the obtained results. In case the reviewer wishes to have more written information on this topic, we are of course willing to re-adjust this part of the manuscript

  • Adjusted text/table/figure: In table 2, the prevalence of NCD behavioural risk factors are presented for all students and for the different years. No significant differences were detected in these health behaviours between academic years, and high prevalence ratios were found for poor sleep quality, low levels of habitual physical activity and low-to-moderate diet quality across the different stages of the degree program. Indicators of smoking status and alcohol consumption were excellent, with prevalences ranging between 2.1% and 6.9% for all students and each academic year. A low to moderate diet quality seemed to decrease slightly during the medical education program, however, no significant difference between academic years was observed.

In table 3, the prevalence of students with no, one and multiple risk factors is presented. Of the medical students that participated in this study, a staggering proportion of 62.0% had two or more behavioural risk factors. Surprisingly, only 11.7% of the student sample didn’t possess any behavioural risk factor while 26.3% of this cohort possessed only one risk factor. There were no significant differences in the clustering of behavioural risk factors between students in different academic years of the degree program (p = 0.460).

  1. Interpretation of Results: Discuss what the observed prevalence rates and differences might imply about the impact of medical education on lifestyle habits.

  • Response to the reviewer: We already included a short part on the interpretation of these results into the discussion part, however, we agree with the reviewer that this is indeed too important to discuss too shortly. The idea of the author team was to include this discussion item into the implications for future research which could be found at the end of the conclusion section. Thanks to the reviewers’ comment, we have increased this paragraph in the discussion part, as can be seen below.

  • Adjusted text/table/figure: Despite augmented health literacy, our cohort of medical students was found to possess an unhealthy lifestyle across all years in a South African medical degree program. Moreover, the acquired health literacy did not proportionally change the students’ health behaviour throughout the medical program as no significant differences were found between the different year students. This concerning fact could be due to a lack of preventive or lifestyle healthcare courses or to a lack of time to prepare healthy food, be physically active and sleep well throughout the medical curriculum.

  1. Contextual Relevance: Compare these findings with existing literature on medical students in other regions or countries. This can help to contextualize your results and might suggest areas where the curriculum could be improved to address these risk factors.

  • Response to the reviewer: Notwithstanding the fact that we sincerely understand the reviewers’ question/ remark, it is difficult to meet this requirement as most of the studies published in this field have investigated single years or NCD risk factors. To the best of our knowledge, we have found one Italian study that used a similar research methodology studying multiple risk factors throughout all years of a medical program. In an attempt to meet the reviewers’ requirement, we have added a small part to this paragraph to better describe similarities/differences with the latter mentioned study.

  • Adjusted text/table/figure: These results are in agreement with a recent Italian study that presented either analogous or worse health behaviours through a comparable academic program [31]. While a higher proportion of students with multiple NCD risk factors was observed in this current investigation compared to Faggiano et al.[31], it seems that substance use (alcohol and tobacco) was better compared to the Italian medical student sample. Additionally, a similar proportion of students were found to have a poor level of daily physical activity in the South African or Italian medical cohorts. The latter study however, found their cohort to have significantly better nutritional behaviours in the senior years which was not found in our study. It is difficult to compare our findings with other scientific data because other studies either focused on one behavioural risk factor or a certain academic year, according to a recently published meta-analysis[17].

  1. Limitations: Address potential biases or limitations in your study, such as the self-reported nature of the data, which may lead to underreporting or overreporting of certain behaviors.

  • Response to the reviewer: We thank the reviewer for this remark as we forgot to include this important limitation. Hence, we have adjusted this specific paragraph in the text file and added the limitation of self-reported data beyond the other three limitations we described in the manuscript file. The original plan of this study, with measurements of anthropometric and physical testing data, could not be carried out due to the strict COVID lockdown.

  • Adjusted text/table/figure: A first limitation to this study is that response ratios differed significantly between the different years. Although normality was checked and preserved throughout all analyses, this might have had an impact on the analysis of variances between the different years. Also, it was unexpected that COVID-19 would occur and have an impact on the lifestyles and behaviours of medical students. Hence, it was important to take this into account when analyzing the results of this study and could act as both a strength and a limitation of this study. A second limitation was the absence of data of the 4th A third limitation was the self-reported nature of the retrieved data in the current study which has the potential disadvantage of over- or underrepresenting certain lifestyle behaviours. An important strength of this study is the large total sample size of medical students that completed all questionnaires which provides reliable data with regard to the health behaviour of the total sample.

  1. Implications for Medical Education: Based on your findings, recommend specific changes or interventions in the medical curriculum that could help in cultivating healthier lifestyle habits among students.

  • Response to the reviewer: We understand the reviewers’ comment but we think we already sufficiently included certain recommendations in the concluding paragraph. To proof this point we included the specific paragraph below. If the reviewer wants to see an additional intervention added to this part, we are of course open to listen to more specific recommendations.

  • Non- adjusted text/table/figure: Based on these results, we recommend that future interventions implemented at tertiary institutions could include: 1. Strengthening the medical and allied health curricula to include more education about NCDs, their risk factors and the prevention thereof, including areas, such as behaviour change theory, to provide a deeper understanding of the topic. 2. Reimagining the built environment to facilitate changes in behaviour e.g. healthy food options at various canteens/cafeteria; indoor/outdoor exercise facilities available to students; safe outdoor spaces/green corridors to encourage walking and meeting on campus, building design that encourages physical activity. 3. Campus-wide health promotion initiatives including communications, posters and activations to create awareness.

  1. Future Research: Suggest areas for further research, such as longitudinal studies to track changes in student behaviors over time or intervention studies to test specific changes in curriculum or support services.

  • Response to the reviewer: We agree with the reviewers’ point of view to additionally include a perspective for future research as this was not present in the current first version. Hence we have added a new section at the end of the manuscript with some future perspectives.

  • Non- adjusted text/table/figure: Taking into consideration the widespread effects on health, future research should focus on solving this important issue by longitudinal evaluations of these risk factors before and after certain implemented initiatives during the academic program such as an augmented amount of specific and NCD preventive oriented courses, some movement classes or active breaks and the availability of affordable, healthy food for students.

Reviewer 2 Report

Comments and Suggestions for Authors

Dear Authors,

the manuscript collected data of medical students and presented the prevalence of risk factors for the NCD.

I have some comments as follows:

- abstract: the statistical analysis description can be taken away. 'Differences in preva- 27
lence of risk factors between academic years were assessed using a Chi square test, and differences 28
for categorical modifiable risk factors were assessed using one-way ANOVA (significance = p<0.05).'

Introduction:

- starting the introduction with the NCD prevalence sound a bit out of scope with the aim of the study. I suggest beginning with medical students/ or healthcare professional lifestyles or risks or prevalence of diseases related to medical professionals works.

Method: Since medical students are vulnerable group. Have they been explained that the research and responses would not impact their learning schedules/outcomes/ scores. Please elaborate more on how the consent was collected, how the research method  were explained to the students.

- what are the response rate, it was a bit strange that all included answered the surveys.

- what were the effect/sample size for this research?

- please describe the context of the medical school in South Africa. Is it 6 years curriculum.

Results: regression analysis would be interesting to see the trend of associations.

Discussion: you have made very interesting points.

Reference: some of the references were outdated.

Comments on the Quality of English Language

can be improved

Author Response

Response to the reviewer (Reviewer 2)

Overall response to the reviewer

  • The author team wants to thank the reviewer sincerely for taking the time to read through the manuscript thoroughly and giving interesting, specific advices or detect some mistakes that were still present in the text. We are aware that this takes a lot of time and appreciate the effort. We hope the level of the manuscript increased and is now acceptable for the reviewer.

Specific comment responses

  1. abstract: the statistical analysis description can be taken away. 'Differences in prevalence of risk factors between academic years were assessed using a Chi square test, and differences for categorical modifiable risk factors were assessed using one-way ANOVA (significance = p<0.05).'

  • Response to the reviewers’ question/remark: We thank the reviewer for his or her comments with regard to the abstract because it might extend this summary unnecessarily. For this reason, we have left out this part in the manuscript.

  • Added/Adjusted abstract: Abstract: Background: Personal health behaviours and lifestyle habits of health professionals influence their counselling practices related to non-communicable diseases (NCDs). There are limited data on the prevalence of unhealthy lifestyle habits among medical students and the impact of acquired health knowledge throughout the curriculum. To determine and compare the prevalence of modifiable behavioural NCD risk factors of medical students in different academic years at a South African tertiary institution. Methods: A cross-sectional observational study of 532 consenting medical students. Participants completed five online questionnaires regarding lifestyle behaviours (physical activity, dietary habits, smoking, alcohol consumption and sleep). Results: Lifestyle-related risk factors with the highest prevalence were: poor sleep quality (66.0%), low levels of habitual physical activity (55.8%) and low-to-moderate diet quality (54.5%). There were no differences between academic years for all risk factors measured. Over 60% of the cohort had 2 or more NCD risk factors and this prevalence did not differ across the degree program with the acquisition of more health knowledge. Conclusion: Medical students have a high prevalence of poor sleep quality, low levels of physical activity and low-to-moderate diet quality, which does not appear to change over the course of their academic career. Sleep hygiene, regular physical activity and healthy nutrition should be targeted in intervention programmes and be more prevalent in the medical curriculum.

  1. Introduction: starting the introduction with the NCD prevalence sound a bit out of scope with the aim of the study. I suggest beginning with medical students/ or healthcare professional lifestyles or risks or prevalence of diseases related to medical professionals works.
  • Response to the reviewers’ question/remark: This was in fact a very interesting remark, made by the reviewer because the initial part of the introduction does not start with the core of the current study. We are, for that reason, very grateful for the reviewers’ remark and adjusted the first part of the introduction accordingly.

  • Added/Adjusted text: Personal health behaviours of health professionals have been presented as im-portant determinants on both their counselling practices and the community health behaviour and health in general [1-6]. For example, non-smoking medical doctors who engage in regular physical activity, have good nutritional habits and a healthy body composition are more likely to prescribe lifestyle behavioural changes and are perceived as more credible by their patients in general [1-6]. A healthy lifestyle behaviour has been repeatedly mentioned as the most important countermeasure in order to prevent Non-Communicable Diseases (NCDs) and for that reason, it is crucial to have medical doctors on board as they are considered as the representatives of ’health’ by their direct environment [7-9]. NCDs are the leading cause of death and morbidity globally, being responsible for approximately 70% of all deaths[7-9]. While the World Health Organi-sation (WHO) has attributed 51% of South African deaths to NCDs, it is clear that people of all age groups, regions and countries are vulnerable to the risk factors contributing to NCDs. Physical inactivity, unhealthy diet, tobacco use and the harmful use of alcohol, are regarded as the most important modifiable behavioural contributors to the devel-opment of NCDs[7-9]. Additionally, a considerable body of epidemiological evidence adds poor sleep to that list of crucial behavioural NCD risk factors[10,11].While there is an abundance of literature focusing on the prevalence of a single behavioural Non-Communicable Disease (NCD) risk factor in (medical) students, data about com-bined NCD risk factors is surprisingly scarce, especially from the African continent [12-17]. Taking into account the important impact of the health professionals’ (health) behaviour for NCD treatment and prevention of the broader community and the scarcity of data on this issue in the South African context, it is crucial to investigate lifestyle habits during the student period as it seems that these habits, established in the univer-sity student period, will determine health behaviour later in life and significantly im-prove academic performance as well as NCD prevention and treatment of the broader community [12-16].

  1. Method: Since medical students are vulnerable group. Have they been explained that the research and responses would not impact their learning schedules/outcomes/ scores. Please elaborate more on how the consent was collected, how the research method  were explained to the students.

  • Response to the reviewers’ question/remark: A member of the research team arranged an information session with the relevant students, during which the research project and procedures were explained to all potential participants. They were given the opportunity to ask questions, were provided with a Participant Information sheet, and consent was taken in English with each questionnaire. It was explained to prospective participants that their participation in this study was entirely voluntary, that they could refuse to participate or withdraw at any time without stating any reason, and that their withdrawal from the research would not affect the training and information they would gain as part of their curriculum.  It was further explained that all information obtained during the course of the study would be regarded as confidential and results of the research will be published or presented in such a fashion that participants remain unidentifiable. We agree with the reviewer that this point could be more thoroughly explained in the method section so we adjusted this part accordingly.

  • Added/Adjusted text: In a cross-sectional observational study conducted in 2020, medical students of 18 years or older and currently registered for a Bachelor of Medicine and Surgery (MBChB, a 6-year medical degree) at the Faculty of Health Sciences, University of Pretoria were invited to complete five online questionnaires. After an information session at which the research project and procedures were explained to all potential participants, 532 medical students from the 1st,2nd, 3rd and 5th year (44.6% of the invited sample) gave their consent and completed five online questionnaires via the online Smartabase plat-form/application (Fusion Sports Pty(Ltd)). The cohort that completed all questionnaires consisted out of 356 females (66.9%) and 176 males (33.1%) and the age range of our sample is between 20-24 years old.

  1. what are the response rate, it was a bit strange that all included answered the surveys.

  • Response to the reviewers’ question/remark: We agree with the reviewer that this information was not sufficiently described and for that reason we have added the response rate to the manuscript file. In total, 44.6% of the invited sample, gave their consent and completed five online questionnaires.

  • Added/Adjusted text: Participants and Procedures: In a cross-sectional observational study conducted in 2020, 532 medical students from the 1st,2nd, 3rd and 5th year (44.6% of the invited sample) gave their consent and completed five online questionnaires via the online Smartabase platform/application (Fusion Sports Pty(Ltd)).

  1. - what were the effect/sample size for this research?

  • Response to the reviewers’ question/remark: The reviewers’ remark about the sample size has been handled with other remarks from this and other reviewers who requested more information about the procedure to get this sample size and the proportion of students that consented for participation in this study. Hence we hope this will meet the reviewers’ requirements. With regard to effect sizes, our responsible statistician is a bit doubtful which exact number the reviewer wants except the p-value. Of course, it is irrelevant to only give a p-value in case of, for example, a logistic regression or a correlation model where the odds ratios and or the Pearson r- value is crucial to add , but in this case we were a bit in doubt which values the reviewer would like to see in the manuscript. Of course we would be happy to carry out eventual adjustments in case the reviewer is not satisfied without the inclusion of the required effect sizes. Below we have included the added text to clarify the participant recruitment process and sample size (Same section as included before).

  • Added/Adjusted text: In a cross-sectional observational study conducted in 2020, medical students of 18 years or older and currently registered for a Bachelor of Medicine and Surgery (MBChB, a 6-year medical degree) at the Faculty of Health Sciences, University of Pretoria were invited to complete five online questionnaires. After an information session at which the research project and procedures were explained to all potential participants, 532 medical students from the 1st,2nd, 3rd and 5th year (44.6% of the invited sample) gave their consent and completed five online questionnaires via the online Smartabase plat-form/application (Fusion Sports Pty(Ltd)). The cohort that completed all questionnaires consisted out of 356 females (66.9%) and 176 males (33.1%) and the age range of our sample is between 20-24 years old.

  1. - please describe the context of the medical school in South Africa. Is it 6 years curriculum.
  • Response to the reviewers’ question/remark: The Bachelor of Medicine and Surgery (MBChB, a 6-year medical degree) at the Faculty of Health Sciences, University of Pretoria is indeed a 6-year medical training. We have included this now in the manuscript file as well and more specifically, in the methods section.

  1. Results: regression analysis would be interesting to see the trend of associations.

  • Response to the reviewers’ question/remark: We thank the reviewer for her/his for the point made here, however, we do not really comprehend how a linear/logistic regression analysis could work in the current statistical model and research aim. We are of course open to discuss or/and adjust our statistical model used, but we are wondering what to use as the dependent variable in such a case. In the current study we seek for the prevalence proportion of risk factors and eventual differences between years, but we don’t really have a measured dependent variable (yi) which could be causally linked with the risk factors we measured such as de development of Diabetes, hyperglycemia, hypertension,... If the reviewer has more specific suggestions about this point, we are of course open to re-consider and listen to the advice.

  1. Discussion: you have made very interesting points.

  • Response to the reviewers’ question/remark: It is nice to hear that the reviewer appreciated this part of the paper.

  1. Reference: some of the references were outdated.

  • Response to the reviewers’ question/remark: We apologize for this and are a bit surprised by this comment as we tried to include up to date papers about, for example, the impact of COVID lockdowns on the perceived risk factors of students worldwide to include some discussion points about this issue. Taking into consideration the positioning of our results compared to other studies, it was difficult to find studies that did a similar study analyzing all risk factors in (almost) all years during a medical program. That is why we referred to the Italian study quite often and were not able to include more up to date papers in this regard. Also, to position our poor sleep results for example, we chose to include meta-analytical data because the quality of this type of ‘collected’ data is always more robust than using data from recent studies. If the reviewer has some suggestions with regard to more relevant references, we are of course willing to include them.

Reviewer 3 Report

Comments and Suggestions for Authors

Thank you for the opportunity to review the article “ Poor Health Behaviour across a Medical Academic Program at a South African University” The health behavior of medical students should be an example for the population. They are the ones that minimize illnesses and lead to healthy and active lifestyles. The paper describes the basic factors affecting the quality of life of medical students in Africa.

However, I have several suggestions that should be clarified and clarified for the value of the work: 

1. the title of the paper needs to be improved in terms of study time rather than the medical program.

2. was the original version of the GATS questionnaire used in the study, or was an adapted version used?

3) And was it adapted to the regulation of anti-tobacco issues in Africa?

4. the results of the study are laconically described - they need to be detailed for the clarity of the paper.

5. verse 167 - what risk factors did the authors have in mind.

6. Verse 169 what 2 abnormal factors predominated in the study groups, did they overlap in the years of study - please clarify.

7. Verse 201 - the discussion should include reliable data and credible references without speculation that may mislead the reader.

8. were students asked about the impact of Covid 19 on health behavior?

Author Response

Response to the reviewer (Reviewer 3)

Overall response to the reviewer

  • The author team wants to thank the reviewer sincerely for taking the time to read through the manuscript thoroughly and giving interesting, specific advices or detect some mistakes that were still present in the text. We are aware that this takes a lot of time and appreciate the effort. We hope the level of the manuscript increased and is now acceptable for the reviewer.

Specific comment responses

  1. the title of the paper needs to be improved in terms of study time rather than the medical program.

  • Response to the reviewers’ question/remark: We thank the reviewer for his or her comments with regard to the title as this is always a crucial part of the paper. Hence, we hope the reviewer will sufficiently appreciate the updated title.

  • Added/Adjusted title: Poor Health Behaviour does not improve during an Academic Medical Training at a South African University

  1. was the original version of the GATS questionnaire used in the study, or was an adapted version used?

  • Response to the reviewers’ question/remark: We used an adapted version of this questionnaire with the most important outcome if the students smoked in the past or/and the present. This was a very conscious choice by the researchers as the odds for student drop out, due to the multitude of questionnaires that had to be completed by the students, could have been too substantial. We hope the reviewer understands the reason for the fact we used a reduced version of this questionnaire

  1. And was it adapted to the regulation of anti-tobacco issues in Africa?

  • Response to the reviewers’ question/remark: We apologize to the reviewer, but our research team did not take this into account in our study design.

  1. the results of the study are laconically described - they need to be detailed for the clarity of the paper.

  • Response to the reviewers’ question/remark: We thank the reviewer for this remark and apologized for the laconic approach in the pre-revised manuscript. To meet the reviewers’ requirement we revised the written part of this section to create a more elaborated results part. We hope this provides now a more detailed overview of the most important results.

  • Added/Adjusted text: In table 1, the average total scores for questionnaires (where a score is calculated), are depicted for all the students and for the different academic years respectively. Although no significant differences were observed between the academic years for these health behavioural risk factors, it seems that physical activity increased slightly over the degree program (p=0.219). On average, the Total Kasari FIT Index score for all students and for each year was slightly above cut-off criterion of <36 (indicating low habitual physical activity) thus on average the student base reported moderate levels of physical activity, although the variance around these averages was large. The total PSQI score for all students, and for each academic year, was consistently above the cut-off criterion of 5 or more, indicating overall poor sleep quality. Also, the average diet quality was in the moderate range of 20-29 for all students and for each academic year, except the 5th In the last year of their academic program, the nutritional quality was slightly higher than 29. The average AUDIT score was considerably lower (<8) which is indicative for a good average attitude towards alcohol in the current cohort.

Table 1. Average health behaviour profiles (means ±SD) of first, second, third- and fifth-year medical students.

All participants

1st year (n=187)

2nd year (n=232)

3rd year (n=43)

5th year (n=70)

p-values

Mean ± SD

Sleep quality (PSQI score)

5.92 ± 2.93

5.74 ± 2.73

6.09 ± 2.86

6.05 ± 2.76

5.74 ± 3.69

0.624

Kasari FIT Index score

38.73 ± 27.37

37.33 ± 27.45

37.58 ± 27.73

41.88 ± 26.02

44.36 ± 26.47

0.219

REAP-S score

28.68 ± 4.46

28.73 ± 4.19

28.43 ± 4.61

28.98 ± 5.09

29.23 ± 4.31

0.576

AUDIT score

1.87 ± 2.40

1.63 ± 2.07

2.05 ± 2.76

1.81 ± 2.10

1.93 ± 2.11

0.343

* significant difference (p < 0.05).

In table 2, the prevalence of NCD behavioural risk factors are presented for all students and for the different years. No significant differences were detected in these health behaviours between academic years, and high prevalence ratios were found for poor sleep quality, low levels of habitual physical activity and low-to-moderate diet quality across the different stages of the degree program. Indicators of smoking status and alcohol consumption were excellent, with prevalences ranging between 2.1% and 6.9% for all students and each academic year. A low to moderate diet quality seemed to decrease slightly during the medical education program, however, no significant difference between academic years was observed.

Table 2. Prevalence (%) of modifiable NCD-related risk factors in 1st year, 2nd year, 3rd year and 5th year medical students.

All

1st years

2nd years

3rd years

5th years

p-values

N

%

N

%

N

%

N

%

N

%

Past or current smoking

29

5.5

7

3.7

16

6.9

2

4.7

4

5.7

0.560

Harmful alcohol use (AUDIT score >=8)

22

4.1

4

2.1

13

5.6

1

2.3

4

5.7

0.260

Low physical activity (FIT index <=36)

297

55.8

110

58.8

131

56.5

21

48.8

35

50.0

0.469

Moderate physical activity (FIT index >36 - <64)

103

19.4

36

19.3

42

18.1

11

25.6

14

20.0

0.724

Low Frequency (<3 times/week)

128

24.1

43

23.0

53

22.8

14

32.6

18

25.7

0.550

Low to moderate intensity

290

54.5

107

57.2

128

55.2

20

46.5

35

50.0

0.517

Time (<30 minutes/ session)

267

50.2

99

52.9

120

51.7

17

39.5

31

44.3

0.294

Low to moderate diet quality (REAP-S score <=29)

290

54.5

103

55.1

134

57.8

22

51.2

31

44.3

0.245

Low diet quality (13-19)

17

3.2

5

2.7

9

3.9

2

4.7

1

1.4

0.679

Moderate diet quality (20-29)

273

51.3

98

52.4

125

53.9

20

46.5

30

42.9

0.376

Poor sleep quality (PSQI score >5)

351

66.0

125

66.8

157

67.8

30

69.8

39

55.8

0.270

* significant difference (p<0.05).

In table 3, the prevalence of students with no, one and multiple risk factors is presented. Of the medical students that participated in this study, a staggering proportion of 62.0% had two or more behavioural risk factors. Surprisingly, only 11.7% of the student sample didn’t possess any behavioural risk factor while 26.3% of this cohort possessed only one risk factor. There were no significant differences in the clustering of behavioural risk factors between students in different academic years of the degree program (p = 0.460).

Table 3. Prevalence (%) of multiple modifiable NCD-related risk factors in 1st year, 2nd year, 3rd year and 5th year medical students. (p=0.460)

1st years

2nd years

3rd years

5th years

All

N

%

N

%

N

%

N

%

N

%

No risk factors

24

12.8

21

9

5

11.6

12

17.1

62

11.7

One risk factor

43

23

64

27.6

13

30.2

20

28.6

140

26.3

Two or more risk factors

120

64.2

147

63.4

25

58.2

38

55.3

330

62

  1. verse 167 - what risk factors did the authors have in mind.

  • Response to the reviewers’ question/remark: We are grateful for this remark because there might have been confusion by the used word ‘prominent risk factors’. The important message we wanted to disseminate in the first sentences of the discussion part was that poor sleep quality (66.0%), physical inactivity (55.8%) and low to moderate diet quality (54.5%) were the most prevalent risk factors in our sample. So, it is these risk factors we have in mind because they were most prevalent. To solve this confusing sentence, we have adjusted the text accordingly.

  • Added/Adjusted text: The first finding was a surprisingly high prevalence of behavioural NCD-related risk factors in medical students through all years of the medical degree program, with poor sleep quality (66.0%), physical inactivity (55.8%) and low to moderate diet quality (54.5%) being the most predominant risk factors.

  1. Verse 169 what 2 abnormal factors predominated in the study groups, did they overlap in the years of study - please clarify.

  • Response to the reviewers’ question/remark: Again, we think this sentence might have been written in a confusing way and we thank the reviewer to reveal this issue via her/his comments. As pointed out in the previous response, poor sleep quality, physical inactivity and low to moderate diet quality were the most prevalent risk factors in our sample. In the referred sentence here, we wanted to present other results we calculated with regard to the clustering of different risk factors. In this case, we did not calculate which risk factors clustered specifically, but rather wanted to demonstrate that a large amount of students possessed 2 or more (variable) risk factors throughout all years of their medical training. This is relevant to report because the clustering of NCD risk factors increases, of course, the risk to develop chronic diseases in the long term. We have adjusted the text slightly so this point could be clearly made.

  • Added/Adjusted text: Secondly, over 60% of the medical students were found to have 2 or more of these modifiable risk factors, and the prevalence of these clustered risk factors (2 or more) was not significantly different between students in different years of their academic career.

  1. Verse 201 - the discussion should include reliable data and credible references without speculation that may mislead the reader.

  • Response to the reviewers’ question/remark: We are unsure about the reason why the reviewer thinks that this reference was not credible for the statement provided here, but we removed the sentence as it was not key in this discussion part. It did not contain any supplementary or crucial information with regard to poor sleep, physical inactivity or bad nutritional quality. We hope the reviewer is satisfied with the removal of this sentence/reference.

  1. were students asked about the impact of Covid 19 on health behavior?

  • Response to the reviewers’ question/remark: This is a very relevant question and the short answer is no. The reason for this was the fact that this study was already set up before and adjusted to the COVID lockdown situation with the online collection of questionnaire data via the Smartabase platform and the cancellation of the physical measurements.

Reviewer 4 Report

Comments and Suggestions for Authors

Dear Editor,

Thank you for the opportunity to review this manuscript. Below are additional comments and suggestions for this manuscript.

  1. In the abstract, please write “Aim” before the sentence “to determine …”
  2. Method: Please explain how to calculate the sample size, inclusion criteria, and exclusion criteria of participants.
  3. Why did fourth-year students not consent to participate in the study?
  4. Questionnaires: Did authors obtain permission to use the questionnaires from the original authors for their research?
  5. Results: On page 4, lines 156–157, “Of the medical students that participated in this study, 64.2% had two or more behavioural risk factors.” In table 3, the percentage of 64.2% for two or more risk factors is the value for the first-year students, while the percentage for the total participants is 62%. Please clarify these values. Please also check the values for the no-risk factor group and the one-risk factor group between the explanation on lines 157–158 and those in Table 3.

Author Response

Response to the reviewer (Reviewer 4)

Overall response to the reviewer

  • The author team wants to thank the reviewer sincerely for taking the time to read through the manuscript thoroughly and giving interesting, specific advices or detect some mistakes that were still present in the text. We are aware that this takes a lot of time and appreciate the effort. We hope the level of the manuscript increased and is now acceptable for the reviewer.

Specific comment responses

  1. In the abstract, please write “Aim” before the sentence “to determine …”

  • Response to the reviewer: We thank the reviewer for her/his sharp eye, because the sentence used in the abstract was not correct. As the reviewer will see, this sentence has been adjusted accordingly.

  • Adjusted/Added text: The aim of this study was to determine and compare the prevalence of modifiable behavioural NCD risk factors of medical students in different academic years at a South African tertiary institution.

  1. Method: Please explain how to calculate the sample size, inclusion criteria, and exclusion criteria of participants.

  • Response to the reviewer: Students 18 years or older and currently registered for a Bachelor of Medicine and Surgery (MBChB, a 6-year medical degree) at the Faculty of Health Sciences, University of Pretoria were included in the study.  Medical students who did not consent for their data to be used for research purposes were excluded from the study. The included sample size is the amount of students that agreed to give their consent in order to use the questionnaire data for research purposes. 44.6% of the invited sample gave their consent and completed five online questionnaires via the online Smartabase platform/application (Fusion Sports Pty(Ltd)). In order to create more clarity about the inclusion process and criteria, we have adjusted the initial part of the methods section. By doing so, we hope the reviewer has now, a more profound insight into the used procedures and methods.

  • Adjusted/Added text: In a cross-sectional observational study conducted in 2020, medical students of 18 years or older and currently registered for a Bachelor of Medicine and Surgery (MBChB, a 6-year medical degree) at the Faculty of Health Sciences, University of Pretoria, were invited to complete five online questionnaires. After an information session at which the research project and procedures were explained to all potential participants, 532 medical students from the 1st,2nd, 3rd and 5th year (6% of the invited sample) gave their consent and completed five online questionnaires via the online Smartabase platform/application (Fusion Sports Pty(Ltd)). The cohort that completed all questionnaires consisted out of 356 females (66.9%) and 176 males (33.1%) and the age range of our sample is between 20-24 years old.

  1. Why did fourth-year students not consent to participate in the study?

  • Response to the reviewer: We thank the reviewer for this question, because this was annoying for the research team as well. As mentioned before, It was explained to prospective participants that participation in this study was entirely voluntary, that they could refuse to participate or withdraw at any time without stating any reason. However, we do not have a clear idea why the fourth years collectively neglected to participate as they did not ostentatiously signaled resistance to this project during the information session we provided. A potential reason could be that they were already dissatisfied with their busy calendar in that particular year and that this could be viewed as a sort of resistance act by their side.

  1. Questionnaires: Did authors obtain permission to use the questionnaires from the original authors for their research?

  • Response to the reviewer: All the validated questionnaires used in this study were available/allowed to use for scientific research purposes without any additional charges.

  1. Results: On page 4, lines 156–157, “Of the medical students that participated in this study, 64.2% had two or more behavioural risk factors.” In table 3, the percentage of 64.2% for two or more risk factors is the value for the first-year students, while the percentage for the total participants is 62%. Please clarify these values. Please also check the values for the no-risk factor group and the one-risk factor group between the explanation on lines 157–158 and those in Table 3.

  • Response to the reviewer: We thank the reviewer for this remark as this was a clear mistake of the authors with an included discrepancy in the manuscript between the numbers explained in the text and in the Table. It is definitely 62% of the students that had 2 or more risk factors clustered. Hence, we adjusted the text accordingly and it should now be in agreement with the numbers included in Table 3.

  • Adjusted/Added text: In table 3, the prevalence of students with no, one and multiple risk factors is presented. Of the medical students that participated in this study, a staggering proportion of 62.0% had two or more behavioural risk factors. Surprisingly, only 11.7% of the student sample didn’t possess any behavioural risk factor while 26.3% of this cohort possessed only one risk factor. There were no significant differences in the clustering of behavioural risk factors between students in different academic years of the degree program (p = 0.460).

Table 3. Prevalence (%) of multiple modifiable NCD-related risk factors in 1st year, 2nd year, 3rd year and 5th year medical students. (p=0.460)

1st years

2nd years

3rd years

5th years

All

N

%

N

%

N

%

N

%

N

%

No risk factors

24

12.8

21

9.1

5

11.6

12

17.1

62

11.7

One risk factor

43

23.0

64

27.6

13

30.2

20

28.6

140

26.3

Two or more risk factors

120

64.2

147

63.4

25

58.1

38

54.3

330

62.0

Round 2

Reviewer 1 Report

Comments and Suggestions for Authors

The design of study and graph must br added. I agree with another revisor. Thanks a lot

Comments on the Quality of English Language

Must be improve.

Author Response

Response to the reviewer (Reviewer 1)

  • Overall response to the reviewer: The author team wants to thank the reviewer sincerely for taking the time to read through the manuscript a second time. We hope the adjusted manuscript now fulfills the needs of the reviewer.

Specific comment responses

  1. The design of study and graph must br added. I agree with another revisor. Thanks a lot.

  • Response to the reviewers’ question/remark: We have applied some major changes to the current manuscript with regard to the design of the study. Also, we have even included the study design in the methodology of this study. Hence, we hope these adjustments will suffice to meet the reviewer requirements. With regard to the graph that should be added, we are unsure about which graph should be added, according to the reviewer. In case we need to add specific graphical support, we would be able to re-consider this issue, raised by the reviewer.

  • Adjusted title: Poor Health Behaviour in medical students at a South African University: a cross-sectional survey Study.

  • Adjusted text study design/methodology: In a cross-sectional observational study conducted in 2020, medical students of 18 years or older and currently registered for a Bachelor of Medicine and Surgery (MBChB, a 6-year medical degree) at the Faculty of Health Sciences, University of Pretoria, were invited to complete five online questionnaires. After an information session at which the research project and procedures were explained to all potential participants, 532 medical students from the 1st,2nd, 3rd and 5th year (6% of the invited sample) gave their consent and completed five online questionnaires via the online Smartabase platform/application (Fusion Sports Pty(Ltd)). Also, it was communicated to all students that no consequences on their examinations and grading would follow in case students did not reply to the surveys. The cohort that completed all questionnaires consisted out of 356 females (66.9%) and 176 males (33.1%) and the age range of our sample is between 20-24 years old.

Reviewer 2 Report

Comments and Suggestions for Authors

Dear Authors,

Thank you for revising the manuscript according to the comments.

I still think that the title did not suit the content well: may be the authors can add something about the method of the study for example:

Poor Health Behaviour in medical students at a South African University: a cross-sectional survey Study.

since number of participants in some years were too low to make a comparison and progression along the year, this phrase should be taken away from the title 'does not improve during an Academic Medical Training'

I would also like the authors to clarify some ethical concerns on the exploration of medical students. In the 'Method and Material' section line 93-95, could you please specify that the students understood that there would be no consequences on their examinations and grading in response or non-response to the survey.

Regarding the effect size, I still think that it is important to calculate and show since your primary outcome was 'the prevalence of modifiable behavioural NCD risk factors in medical students at a South African university'. What were your hypothesis of this study? why collecting prevalence is crucial? should we analyze the association between those modifiable behavioural NCD risk factors and academic years. I believe that calculation around the associations would make the manuscript more interesting. Otherwise this manuscript would seem to be like a descriptive data reporting prevalence of all modifiable behavioural NCD risk factors from medical students.

Discussion:  line 186-187, the authors could not implied the significance of prevalence differences since the numbers of participants for each year were too different, particularly in year 3 and 5. The p-value in Table 2 should not be used to imply anything.

Comments on the Quality of English Language

Minor

Author Response

Response to the reviewer (Reviewer 2)

  • Overall response to the reviewer: The author team wants to thank the reviewer sincerely for taking the time to read through the manuscript thoroughly for the second time and apologize for any inconveniences still present for the reviewer. We hope to be able to provide sufficient additional information and/or responses according to the reviewers’ remarks.

Specific comment responses

Dear Authors,

Thank you for revising the manuscript according to the comments.

  1. I still think that the title did not suit the content well: may be the authors can add something about the method of the study for example: Poor Health Behaviour in medical students at a South African University: a cross-sectional survey Study. since number of participants in some years were too low to make a comparison and progression along the year, this phrase should be taken away from the title 'does not improve during an Academic Medical Training'

  • Response to the reviewers’ question/remark: We agree with the reviewers’ opinion that the title was not adjusted in a correct way as it does/did not reflect the used methodology and analyses we performed in the current study. Hence, we are very grateful to the reviewer to mention this mistake and to provide a valuable alternative.

  • Adjusted title: Poor Health Behaviour in medical students at a South African University: a cross-sectional survey Study.

  1. I would also like the authors to clarify some ethical concerns on the exploration of medical students. In the 'Method and Material' section line 93-95, could you please specify that the students understood that there would be no consequences on their examinations and grading in response or non-response to the survey.

  • Response to the reviewers’ question/remark: To meet the reviewers’ requirements, the authors added the sentence below and acknowledge in all honesty, that all surveys were analyzed by a research group that was not involved in and independent from all teaching activities of the students. Hence, there could not have been an influence on scoring or punishing the students from the research groups’ perspective.

  • Adjusted text: Also, it was communicated to all students that no consequences on their examinations and grading would follow in case students did not reply to the surveys.

  1. Regarding the effect size, I still think that it is important to calculate and show since your primary outcome was 'the prevalence of modifiable behavioural NCD risk factors in medical students at a South African university'. What were your hypothesis of this study? why collecting prevalence is crucial? should we analyze the association between those modifiable behavioural NCD risk factors and academic years. I believe that calculation around the associations would make the manuscript more interesting. Otherwise this manuscript would seem to be like a descriptive data reporting prevalence of all modifiable behavioural NCD risk factors from medical students.

  • Response to the reviewers’ question/remark: We understand the reviewers’ remark, however, we are unsure about the associations the reviewer wants to see in the effect sizes and which specific analysis should be envisaged to study associations between scaled questionnaire responses and a certain academic year. Notwithstanding this unclarity, we have tried to calculate Cohens D effect size for standardized mean differences as it is impossible to perform a correlation or odds ratio (logistic regression) analysis (we don’t have two quantitative markers that could be correlated or a binomial outcome variable to calculate odds ratios for…). These Cohens d effect sizes are -0.08 for the difference between 1st and 2nd year; 0.48 for the difference between the 1st and the 3rd year and 0.162 for the difference between the 1st and the 5th Because these effect sizes do not really reflect associative effect sizes, we decided to not include them into the manuscript. In case of a misunderstanding, we are of course open to hear the reviewers detailed opinion about what statistical technique should be used when envisaging associations between years and risk factors.

  1. Discussion: line 186-187, the authors could not implied the significance of prevalence differences since the numbers of participants for each year were too different, particularly in year 3 and 5. The p-value in Table 2 should not be used to imply anything.

  • Response to the reviewers’ question/remark: We might have expressed the fact that no prevalence differences were found between the different years a bit too strong, and for this reason, we have adjusted the sentence slightly. We hope the reviewer is now satisfied with the sentence below. Because we found no significant p-values, however, between the prevalence p-values in all academic years we still believe we can make the statement, written below.

  • Adjusted text: The prevalence of these risk factors did not differ significantly between the different academic years.
